# Cerebellar folding is initiated by mechanical constraints on a fluid-like layer without a cellular pre-pattern

Andrew K Lawton[1], Tyler Engstrom[2], Daniel Rohrbach[3], Masaaki Omura[3,4,5], Daniel H Turnbull[4], Jonathan Mamou[3], Teng Zhang[6], J M Schwarz[2], Alexandra L Joyner[1,7]*

[1]Developmental Biology Program, Sloan Kettering Institute, New York, United States; [2]Department of Physics, Syracuse University, Syracuse, United States; [3]Lizzi Center for Biomedical Engineering, Riverside Research, New York, United States; [4]Department of Radiology, Skirball Institute of Biomolecular Medicine, NYU School of Medicine, New York, United States; [5]Graduate School of Science and Engineering, Chiba University, Chiba, Japan; [6]Department of Mechanical & Aerospace Engineering, Syracuse University, Syracuse, United States; [7]Biochemistry, Cell and Molecular Biology Program, Weill Graduate School of Medical Sciences, Cornell University, New York, United States

**Abstract** Models based in differential expansion of elastic material, axonal constraints, directed growth, or multi-phasic combinations have been proposed to explain brain folding. However, the cellular and physical processes present during folding have not been defined. We used the murine cerebellum to challenge folding models with in vivo data. We show that at folding initiation differential expansion is created by the outer layer of proliferating progenitors expanding faster than the core. However, the stiffness differential, compressive forces, and emergent thickness variations required by elastic material models are not present. We find that folding occurs without an obvious cellular pre-pattern, that the outer layer expansion is uniform and fluid-like, and that the cerebellum is under radial and circumferential constraints. Lastly, we find that a multi-phase model incorporating differential expansion of a fluid outer layer and radial and circumferential constraints approximates the in vivo shape evolution observed during initiation of cerebellar folding.
DOI: https://doi.org/10.7554/eLife.45019.001

*For correspondence:
joynera@mskcc.org

Competing interests: The authors declare that no competing interests exist.

## Introduction

Recent work to elucidate the mechanics of brain folding has primarily focused on the human cerebral cortex and involved models of directed growth, axonal tension, or differential expansion of elastic materials that generate compressive forces to drive mechanical instabilities leading to folding (*Tallinen et al., 2014*; *Ronan et al., 2014*; *Bayly et al., 2013*; *Xu et al., 2010*; *Hohlfeld and Mahadevan, 2011*; *Bayly et al., 2014*; *Lejeune et al., 2016*; *Karzbrun et al., 2018*). Current elastic material models are able to create three-dimensional shapes strikingly similar to the final folds seen in the adult human cortex (*Tallinen et al., 2016*). A recent multi-phase model (*Engstrom et al., 2018*) that includes elastic and fluid-like layers, differential expansion and radial constraints takes into consideration that multiple factors could lead to folding in the developing brain. However, the cell and tissue level mechanics actually present at the initiation of folding have not been considered or defined, as technological limitations are significant in animals with a folded cerebrum.

**eLife digest** The human brain has a characteristic pattern of ridges and grooves that make up its folded shape. Folds in the outer layer of the brain, known as the cortex, increase the surface area and make more space for cells to connect and form complex circuitries. Different models have been put forward to explain how these folds form during development. Examples include tension from cells pulling areas of the cortex together, or layers of the cortex growing at different rates, causing the cortex to buckle and create folds. Discriminating between these different models requires biological information about the cells and tissue of the brain at the start of the folding process. However, it has been difficult to extract this information when considering the development of the human brain in three dimensions.

Lawton et al. have overcome these difficulties by using a part of the mouse brain called the cerebellum as a simpler system. As in humans, the mouse cerebellum is a densely folded structure, sitting underneath the brain, that plays a major role in regulating movements, as well as cognition. The symmetrical structure of the mouse cerebellum means it can be analyzed in two dimensions, making it easier to track the mechanics of folding.

By applying the extracted biological data onto a mathematical model, Lawton et al. showed folding was driven by a combination of previously unknown features. For instance, that cells in the outer layer of the cerebellum grow faster than cells in the center, with cells growing uniformly across the outer layer. Other features include the fluid-like composition of the outer layer, which allows cells to move freely and regularly change position, and tensions surrounding the cerebellum mechanically straining its growth. Notably, the pattern of cells and tissue fibers in the cortex had no influence over these mechanical properties and provided no pre-indication of where the sites of folding would occur. The data collected deviates from other models, and has led Lawton et al. to propose a new explanation for how the brain folds, incorporating these newly found features.

Problems with brain folding during human development can lead to debilitating conditions. Applying this new model to folding disorders of the human brain could help scientists to understand how these folding defects arise.

DOI: https://doi.org/10.7554/eLife.45019.002

The murine cerebellum has a simple alignment of 8–10 stereotypical folds along the anterior-posterior axis. Combined with the genetic tools available in mouse this allows for precise developmental interrogation to identify and analyze the in vivo cellular and tissue level behaviors driving growth and folding. The developing cerebellum is distinct from the cerebral cortex, as it has a temporary external granule cell layer (EGL) of proliferating granule cell precursors that cover the surface and generate growth primarily in the anterior-posterior (AP) direction (*Leto et al., 2016*; *Legué et al., 2015*; *Legué et al., 2016*). During development a thickening occurs in the EGL at the base of each forming fissure, termed anchoring center (AC) (*Sudarov and Joyner, 2007*), whereas in the adult cerebellum the inner granule cell layer (IGL), generated by the EGL during the first two weeks of life, is thinnest at the ACs. Previous work on cerebellar folding utilized a tri-layer elastic model incorporating the EGL, the adjacent molecular layer, rich in axons and dendrites, and the IGL (*Lejeune et al., 2016*). However, neither the molecular layer nor the IGL are present when folding is initiated in the embryo. Therefore we argue that a bilayer system consisting of the EGL and underlying core, is a more appropriate approximation for cerebellar folding.

Here we show that cerebellar folding emerges from differential expansion between an un-patterned, rapidly expanding EGL and an underlying core. Additionally, we demonstrate that the measured stiffness differential, compressive forces, and the thickness variation in the EGL are all inconsistent with traditional elastic wrinkling models driven by differential growth. Furthermore, we demonstrate that the expansion of the EGL is uniform, and fluid-like, and that the cerebellum is under radial and circumferential constraints when folding initiates. Lastly, we constrain the recent multi-phase model with our in vivo data and find we can capture the temporal shape evolution seen during mouse cerebellum folding initiation. The implications of our findings for human cerebral cortex folding are discussed.

## Results

### Tissue level mechanics drive folding

It is well known that differentially expanding bilayer systems can wrinkle to relax building stress (*Richman et al., 1975*; *Nelson, 2016*; *Hannezo et al., 2012*; *Shyer et al., 2013*; *Wiggs et al., 1997*). We reasoned that in the cerebellum the EGL could behave as a quickly expanding outer layer and its attachment to a more slowly growing core could generate forces that result in a wrinkling-like phenotype. To test whether the cerebellum has differential expansion between the two layers, we measured the expansion of the EGL and the core during the time of initiation of folding from midline sagittal sections (*Figure 1a–d*). Unlike the cerebral cortex, the unfolded murine cerebellum is a simple cylinder-like structure elongated in the medio-lateral axis (*Figure 1e*) (*Szulc et al., 2015*). All folds in the medial cerebellum (vermis) are aligned in the same axis allowing 2-D measurements to estimate expansion in the anterior-posterior axis of the vermis. Therefore the length of the surface of the EGL was used as a measure of the cerebellum surface area and the area of the core as an approximation of cerebellum volume (*Figure 1d*), and measurements were made each day from embryonic day 16.5 (E16.5) through postnatal day 0 (P0). In cross-section the unfolded cerebellum approximates a semicircle, therefore we reasoned that if the cerebellum were to remain unfolded then the ratio of expansion between the length of the EGL and the area of the core should approximate the ratio of the circumference of a semi-circle to its area. Of significance, we found that at E16.5 and E17.5 the ratios of growth between the EGL and core closely approximated the expansion of a semi-circle. However, at E18.5 and P0 the expansion rate of the EGL was greater than the rate of core expansion (*Figure 1f*). Thus we uncovered that the cerebellum does indeed go through a phase of differential expansion. We next determined whether differential expansion correlates with when folding occurs by calculating a folding index (the convex curvature of the EGL divided by the

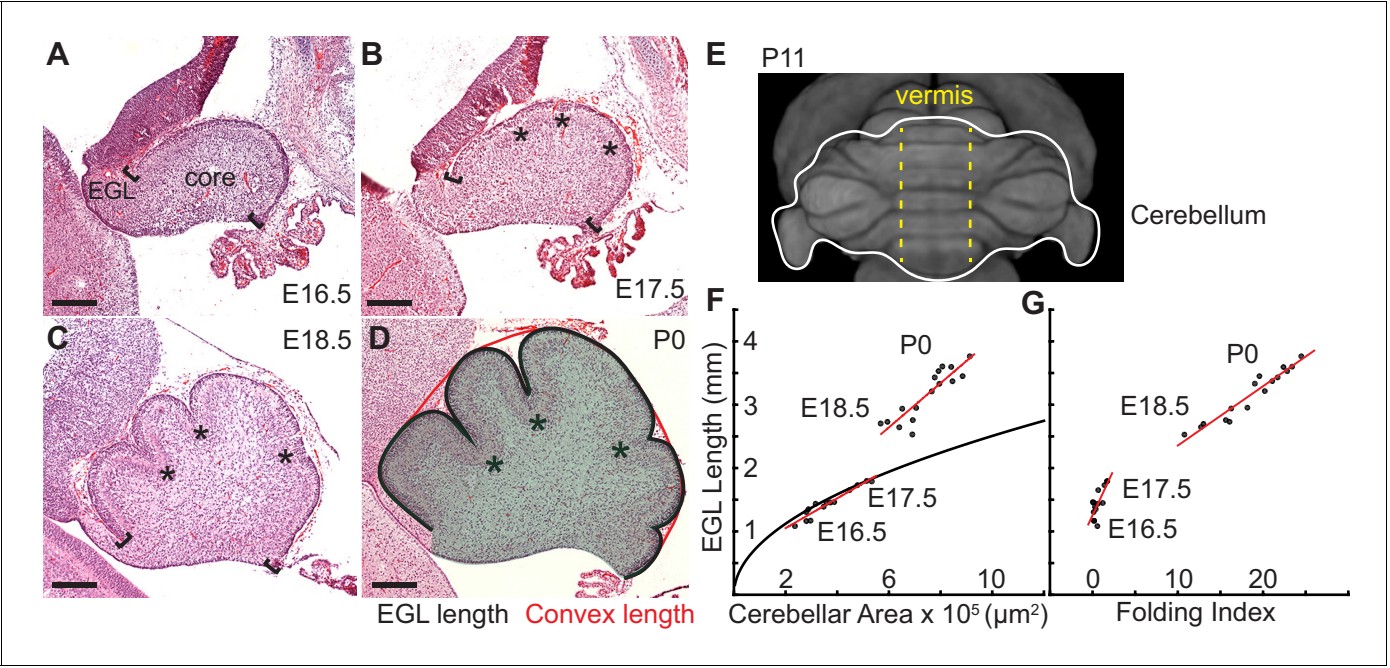

**Figure 1.** Initiation of cerebellar folding correlates with timing of differential expansion. (A–D), H and E stained midline sagittal sections of FVB/N mice at the indicated embryonic (E) and postnatal days (P). Anterior to the left. Stars: ACs. Brackets: anterior/posterior ends of the EGL. Black line and red line in (D): EGL and convex length, respectively. Shaded area: core. (E) manganese enhanced magnetic resonance imaging of P11 cerebellum outlined in white adapted from *Szulc et al. (2015)*. Anterior to the top. Vermis indicated by dotted yellow lines. (F), At E16.5 and E17.5 expansion of EGL length and cerebellar area fit the proportional expansion of a semi-circle (curve). At E18.5 and P0 EGL expansion is greater than core area growth creating differential expansion. (G), Folding index [1 - (convex length/EGL length) x 100] reveals folding initiates during differential expansion. Scale bars: 200 μm.

DOI: https://doi.org/10.7554/eLife.45019.003

length of the EGL) at each stage (*Mota and Herculano-Houzel, 2015*). Indeed, we found that the cerebellum remains unfolded during the initial proportional expansion between the EGL and core and only folds when the differential expansion is initiated (*Figure 1g*). These results provide quantitative evidence that cerebellar folding involves tissue level mechanical forces arising from differential expansion.

## In vivo data contradict elastic bilayer models

Since there is differential expansion between the EGL and the core and as this type of expansion is the driver of elastic bilayer models we tested whether the properties of cerebellar tissue are consistent with the requirements and predictions of such models. Briefly, the initial resulting wrinkling instability defines the distance between folds as the initial sinusoidal undulations increase in amplitude to ultimately turn into lobules. The folding wavelength depends on the thickness of the external layer (EGL) and the ratio of the stiffness of the two layers (EGL/core). In particular, for a planar geometry, with the stiffness of the external layer defined as $E_o$, the stiffness of the core as $E_i$, and the thickness of the external layer denoted as $t$, the folding wavelength $\lambda$ is given by *Allen (1969)*

$$\lambda = 2\,\pi\,t\left(\frac{1}{3}\frac{E_o}{E_i}\right)^{1/3}.$$

If the length of the system is $l$, then the number of folds is inversely proportional to the thickness of the EGL

$$n = \frac{l}{\lambda} \propto \frac{l}{t}\left(\frac{E_i}{E_o}\right)^{1/3}.$$

We explored a standard elastic bilayer model in a circular geometry using the observed ratio of thickness of the EGL to radius of the cerebellum near the onset of shape change (E16.5) and invoked a neo-Hookean elastic solid for both layers (*Zhao and Zhao, 2017*). The resulting shape change was studied as a function of the ratio of the layer stiffness values (*Figure 2a*). We found that to produce the observed number of folds (three in the semi-circular cerebellum and six in the circular model) at initiation of folding through wrinkling based models constrained by our measurements of the embryonic cerebellum, a large stiffness ratio was required of around 50. To map the stiffness contrast in the cerebellum we used scanning acoustic microscopy (SAM) to measure the bulk modulus of the cerebellum daily from E16.5 to P18.5 (*Figure 2b–c*, *Figure 2—figure supplement 1*) using established methods (*Rohrbach et al., 2015*; *Rohrbach et al., 2018*; *Rohrbach and Mamou, 2018*). For small deformations, the instantaneous bulk modulus should linearly relate to the stiffness and, therefore, the ratio of the instantaneous bulk moduli should scale similarly to the ratio of stiffnesses (assuming the same Poisson's ratio for the EGL and for the core, neither of which have been directly measured). While this qualitative approach and SAM tissue preparation protocols may not be able to produce the absolute values of the elastic properties of the tissues, it can give a reasonable indication of the relative stiffnesses of different parts in the cerebellum. Using this estimation, we found that the EGL has a slightly higher instantaneous bulk modulus than the core at all stages measured. Unsurprisingly, the ratio (~1.05:1) was not close to being sufficient to produce a folding wavelength similar to that in the cerebellum (*Figure 2d*). Consistent with our finding, small modulus contrasts have been reported for other brain regions with multiple loading modes, such as shear, compression, and tension (*Xu et al., 2010*; *Lejeune et al., 2016*; *Budday et al., 2017*). Elastic material models with graded growth profiles have been developed that predict folding of cerebral cortex without a large stiffness differential (*Tallinen et al., 2014*). However, these models are still bound by other measurable requirements as discussed below.

Elastic bi-layer wrinkling models predict compressive forces in the outer layer. Simulations performed of cuts through the outer layer and into the inner layer predict that upon relaxation the outer layer should not open (*Figure 2e*). We tested whether this prediction reflects the biology using surgical dissection blades to make radial cuts across the meninges, EGL, and into the core of live E16.5 tissue slices. Time-lapse imaging revealed that, in contrast to the prediction, the EGL opens as well as part of the underlying cut in the core (*Figure 2f–h*, *Figure 2—figure supplement 2a–c*, and *Video 1*). This result indicates there is circumferential tension within the outer layers of the

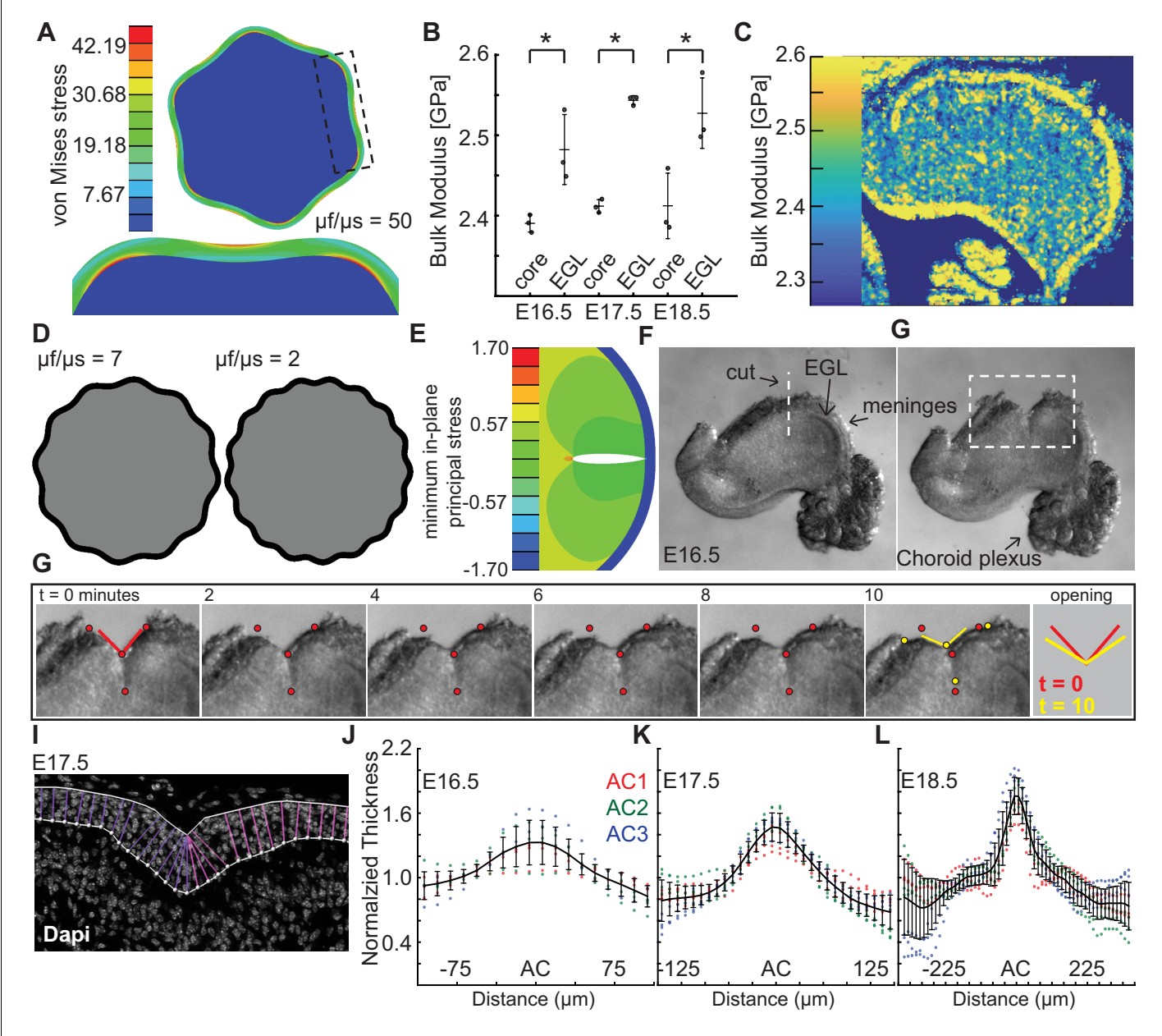

**Figure 2.** Measured tissue stiffness, stress, and shape at folding initiation are inconsistent with wrinkling models. (**A**), Inducing the correct number of folds through a wrinkling model requires a stiffness differential between the layers of 50 fold ($\mu f/\mu s = 50$, g = 1.05). (**B,C**) Acoustic mapping of cerebellar slices show a slightly stiffer EGL than core at each stage (anova df = 5; p=1.0e$^{-4}$, F = 13.59), but not the required differential. Stars indicate statistical differences. (**D**) Wrinkling simulations constrained by developmental data produce wavelengths inconsistent with the embryonic mouse cerebellum. (**E**) Elastic simulations predict the EGL remains closed after cutting. (**F,G**) Images of a live cerebellar slice before and after cutting, and images from time lapse movie, (**H**) show the EGL opens, revealing circumferential tension along the EGL. Red and yellow dots: cut edges. Lines: relaxation angle. (**I**) Staining of nuclei with EGL outlined and lines used to measure thickness. (**J–L**) Normalized EGL thickness (thickness/mean thickness) at the ACs increases during folding initiation (anova E16.5 df = 29, p=8.2e$^{-20}$, F = 12.59. E17.5 df = 29, p=3.4e$^{-116}$, F = 62.78, E18.5 df = 57, p=6.8e$^{-67}$, F = 13.28). At E16.5 only brains with visible ACs were included. Error bars: S.D.

DOI: https://doi.org/10.7554/eLife.45019.004

The following figure supplements are available for figure 2:

**Figure supplement 1.** Examples of the regions measured regions by acoustic microscopy.

DOI: https://doi.org/10.7554/eLife.45019.005

**Figure supplement 2.** The stress patterns within the cerebellum are different between the EGL and the VZ.

DOI: https://doi.org/10.7554/eLife.45019.006

*Figure 2 continued on next page*

*Figure 2 continued*

**Figure supplement 3.** EGL thickness increases in the ACs during the initiation of folding.
DOI: https://doi.org/10.7554/eLife.45019.007

cerebellum. This finding also rules out the elastic models with graded growth profiles as they predict compressive forces in the outer region as well.

The elastic bi-layer model requires the EGL to be thinnest at the base of each AC, which are the lowest parts of the cerebellar surface. Thus, the EGL should have an 'in-phase' thickness variation. Without this feature, a purely elastic model – bi-layer based or even graded growth profile based – cannot be in mechanical equilibrium (in the quasistatic limit) (*Engstrom et al., 2018*). However, we previously reported that the embryonic EGL is thickest in the ACs when folding initiates, that is it has an 'out-of-phase' thickness variation (*Sudarov and Joyner, 2007*). To validate this observation, we quantified the thickness variations in the EGL centered at the ACs present at E16.5–18.5. Not all cerebella have visible AC at E16.5. However in the subset that do and in the three ACs present at E17.5, the EGL was found to be 1.2–1.4 times thicker in the ACs than in the surrounding EGL (*Figure 2i–l* and *Figure 2—figure supplement 3*). Moreover, the thickness ratio increased to 1.7 times at E18.5 (*Figure 2l*). As described above, the final thickness variations of the IGL (as well as the molecular layer) of the cerebellar cortex are in-phase, just as the layers of the adult cerebral cortex. These results further show that traditional elastic wrinkling models cannot capture the initiation of cerebellum folding, and highlight the importance of making biological measurements at the time of folding rather than when it is complete.

## Uniform outer layer expansion without a cellular pre-pattern

As elastic bi-layer models do not align with the biology of cerebellar folding, we looked for other drivers of morphometric changes. Since the EGL drives the majority of cerebellar growth (*Leto et al., 2016*; *Legué et al., 2015*; *Legué et al., 2016*), we first tested whether regional differences in EGL proliferation rates are present that could influence the folding pattern of the cerebellum. Proliferation rates (S phase index) were measured in the EGL during folding initiation (E16.5 and E17.5) in the inbred FVB/N strain to reduce variation between samples. First we asked if the regions that will give rise to distinct sets of lobules have different rates of proliferation that could contribute to the larger and smaller sizes that the lobules ultimately attain. We focused on the anterior cerebellum that divides into a larger region with lobules 1–3 (L123) and smaller region (L45), as well as the central area that comprises lobules 6–8 (L678) of the cerebellum (*Figure 3a–b*). The more posterior cerebellum does not consistently fold at this stage, thus measurements were not included. Interestingly, we found that the proliferation rates were similar in the three regions at E16.5 (*Figure 3c*). The EGL proliferation rate at E17.5 in L678 was slightly reduced compared to the L123 region, but no other differences were found (*Figure 3d*). Thus proliferation is uniform just before initiation of folding and the small difference found during folding does not correlate with lobule size. This result indicates that lobule size is not determined by modulating the levels of proliferation at the onset of folding. Rather, lobule size could be set by both the timing of invagination, and the

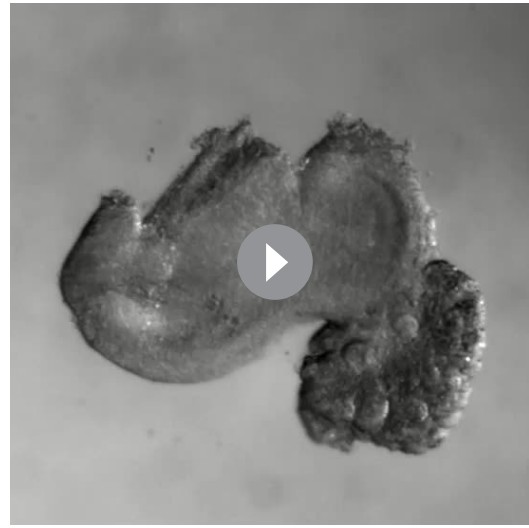

**Video 1.** Live slice cutting and relaxation reveals circumferential tension along the EGL. Time-lapse video shows relaxation of live tissue slice after cutting radially through the EGL and into the underlying core. Images were acquired every 10 s for 10 min. The time-lapse was started moments after the tissue was collected in frame after the cut. The slice shown in the video is the same as in *Figure 2f–h*.
DOI: https://doi.org/10.7554/eLife.45019.008

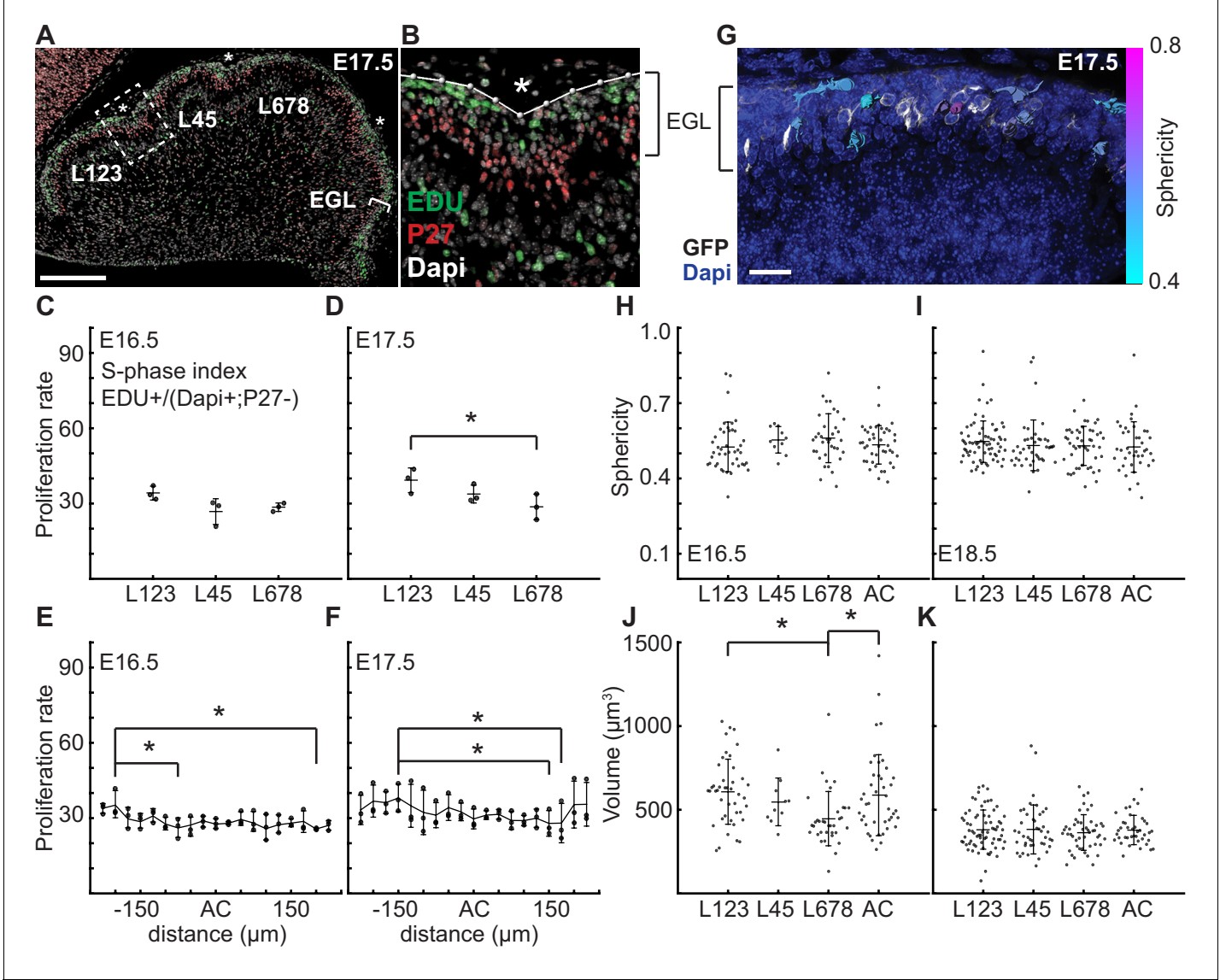

**Figure 3.** At folding initiation the EGL has uniform proliferation, cell size, and cell shape. (A,B) Low and high power images of immuno-histochemical (IHC) staining of sagittal cerebellar sections to measure proliferation in the lobules (L) indicated at 25 μm windows surrounding the ACs (stars). Scale bar: 200 μm. (C,D) EGL proliferation rates are shown before and during the onset of invagination (Two-way anova: df = 2. (C) p=0.10, F = 4.36 (D) p=0.03, F = 10.31). (E,F) Proliferation rates are shown in the AC and in the surrounding EGL showing uniformity (Two-way anova: df = 18. (E) p=0.03, F = 2.15 (F) p=2.1e$^{-3}$, F = 3.06). (G) Section of Atoh1-CreER/+; R26$^{MTMG/+}$ E16.5 cerebellum showing masked labeled cells. Scale bar: 20 μm. (H,I) Cell shape (sphericity) measurements before and during folding (anova df = 3. (H) p=0.34, F = 1.13 i p=0.61, F = 0.61). (J,K) Cell size measurements before and during folding (anova df = 3. (J) p=3.6e$^{-3}$, F = 4.75, (K) p=0.85, F = 0.26). Stars indicate statistical differences. Error bars: S.D.

DOI: https://doi.org/10.7554/eLife.45019.009

The following figure supplement is available for figure 3:

**Figure supplement 1.** Proliferation rate is reduced in the central zone of the cerebellum after folding initiation.

DOI: https://doi.org/10.7554/eLife.45019.010

distance between ACs as granule cell precursors in one lobule do not cross the surrounding ACs to contribute to an adjacent lobule (*Legué et al., 2015*).

Each AC is first detected as a regional inward thickening of the EGL (*Sudarov and Joyner, 2007*) (*Figure 2i–l* and *Figure 2—figure supplement 3*). We measured the proliferation of the EGL specifically within the forming AC regions to test whether altered proliferation rates could explain the thickenings and therefore the initiation of an AC. We found the rate of proliferation within each

forming AC region at E16.5 and E17.5 was the same as in the surrounding EGL (*Figure 3e,f*), thus proliferation within all regions of the EGL at the initiation of folding is uniform. Furthermore, regional modulation of proliferation does not form or position the ACs.

At E18.5, after the initiation of folding, we found that the rate of proliferation was significantly lower in the L678 region compared with the L123 and L45 regions (*Figure 3—figure supplement 1a*). However, proliferation within the ACs at E18.5 remained uniform with the surrounding regions (*Figure 3—figure supplement 1b*). Since ACs compartmentalize the EGL, our results show that regional differences in proliferation rates arise in lobule regions after initiation of folding, which thus could be important for determining the ultimate size of the folds.

Changes in cell size and shape have been shown to induce morphological changes (*Mammoto and Ingber, 2010*; *Harding et al., 2014*; *Stemple, 2005*; *He et al., 2014*). To test if regionally specific regulation of cell shape or size directs folding, we fluorescently labeled cell membranes of scattered granule cell precursors (GCPs) in the EGL using genetics (*Atoh1-CreER/+; R26^{MTMG/+}* mice injected with tamoxifen two days prior to analysis). We then segmented the cells in 3D and quantified their sphericity (*Figure 3g*). We discovered that GCPs in the EGL take on a large variation of shapes and sizes at E16.5 and E18.5. However, we found no difference in cell shape in the different lobule regions of the EGL or between the AC areas and the surrounding EGL at each age (*Figure 3h,i*). Cell size was uniform at both stages except for a slight reduction in L678 at E16.5 when compared with L123 and the AC regions. However, the size of cells is reduced at E18.5 compared to E16.5 (*Figure 3j,k* and *Figure 3—figure supplement 1c*). Thus, the proliferating GCPs that drive expansion of the EGL have both uniform proliferation rates and similar shapes and sizes across the lobule regions defined by the first three ACs at folding initiation.

## Uniform fiber distribution and radial tension at folding initiation

The EGL is traversed by fibers of Bergmann glial and radial glial cells (*Leung and Li, 2018*; *Yuasa, 1996*; *Yamada and Watanabe, 2002*). We tested whether the fibers are distributed in patterns that could locally change the physical properties of the EGL and induce invaginations. Genetics was used to fluorescently label cell membranes of scattered glial cells (*nestin-creER/+;R26^{MTMG/+}* mice injected with tamoxifen at E14.5) (*Figure 4a*). Fibers crossing the EGL at E16.5 were counted in sagittal slices and aligned relative to the ACs (*Figure 4b*). This analysis showed that the Bergmann glial and radial glial fibers are distributed evenly along the AP axis of the EGL, and therefore are not directing the positions where folding initiates based on an uneven regional distribution.

Tension based folding models suggest constraints from axons and other fibers could direct folding (*Xu et al., 2010*; *Van Essen, 1997*). Since the cerebellum is under circumferential tension, as demonstrated above, we examined evidence of radial tension between the EGL and the ventricular zone (VZ) at the initiation of folding. Cuts were made in live E16.5 tissue slices between the EGL and VZ running approximately parallel to them so that they cut across radial fibers in the anterior cerebellum (*Figure 4c*). As predicted, after cutting the tissue relaxed revealing tension directed radially within the cerebellum (*Figure 4d,e* and *Figure 2—figure supplement 2* and *Video 2*). Interestingly, quantification of how the radial and horizontal cuts open revealed that only the horizontal cuts opened along the full length of the cut although they opened more slowly than radial cuts (*Figure 2—figure supplement 2g–j*), indicating different stress profiles in the two orientations.

Taken together, at the time of folding initiation the EGL, which is driving the differential expansion, is itself growing uniformly and the cerebellum is under both radial and circumferential constraints. Finally, there is no evidence of any pre-patterning in the EGL in either cellular behaviors or fiber distribution.

## The EGL is fluid-like as cells undergo dynamic rearrangement

As the granule cells within the EGL have such varied shapes as shown above, we looked to see if the cells within the EGL were undergoing any rearrangement movements that may indicate fluid properties. A small, scattered fraction of nuclei in the EGL were fluorescently labeled (*Atoh1-CreER/+; R26^{ntdTom/+}* injected with tamoxifen two days prior to imaging) and ex vivo slice-culture time-lapse imaging was performed for up to five hours. Tracking the cell positions through time revealed that granule cells within the EGL are highly motile within the EGL. Furthermore, there was no obvious directionality or collectivity to the movement. However, the dynamic motility resulted in the constant

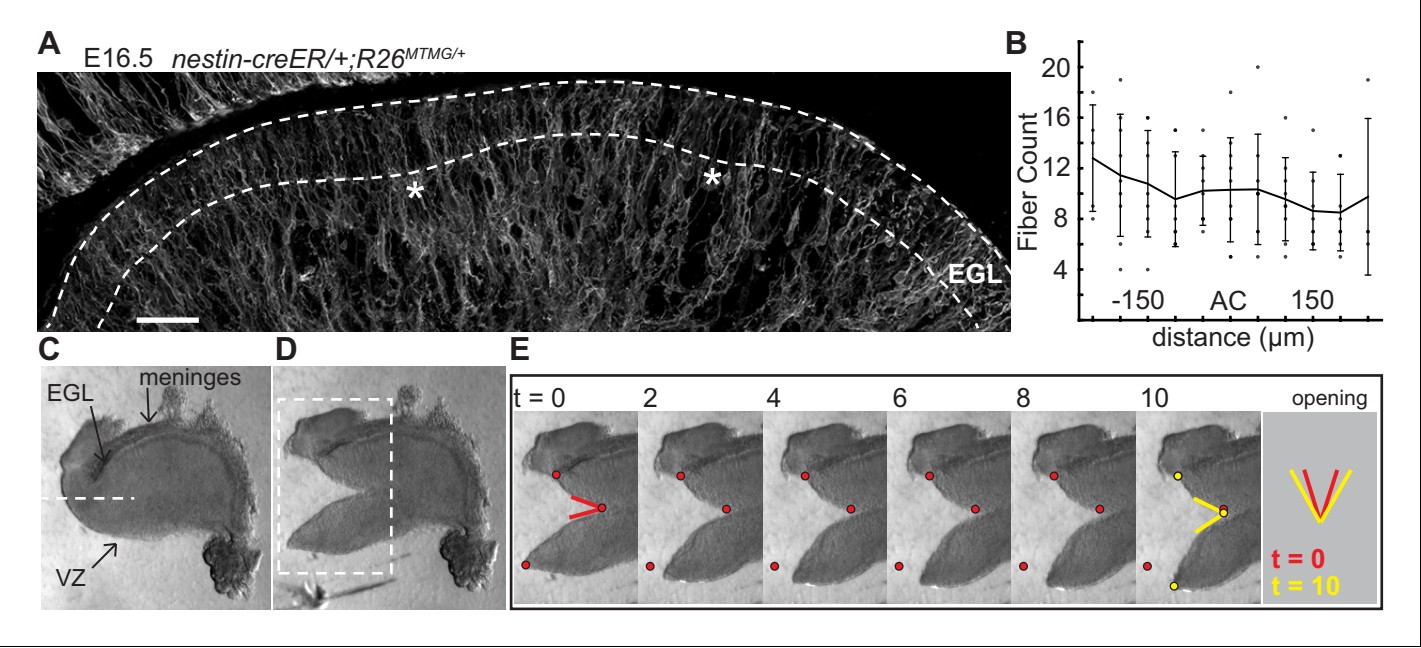

**Figure 4.** The EGL has a uniform distribution of crossing fibers at folding initiation. (**A**) E16.5 nestin-CreER/+; R26[MTMG/+] cerebellum section showing fluorescent labeling of radial and Bergmann glial fibers. Stars: AC. Dotted lines denote EGL. Scale bar: 50 μm. (**B**) Measurements of fiber density in the ACs compared to the surrounding EGL (anova df = 10; p=0.76, F = 0.66). Error bars: S.D. (**C–E**) Still images of a tissue cutting experiment to test for radial tension between the EGL and the VZ. Red and yellow dots: cut edges at t = 0, 10. Lines: relaxation angle.

DOI: https://doi.org/10.7554/eLife.45019.011

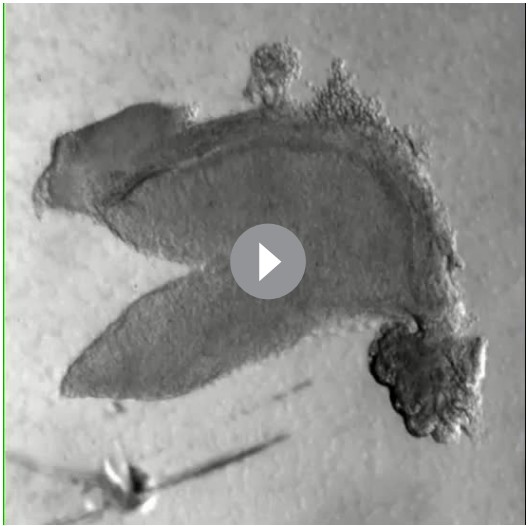

**Video 2.** Live slice cutting and relaxation reveals radial tension between the EGL and The VZ. Time-lapse video shows relaxation of live tissue slice after cutting horizontally into the core between the EGL and the VZ. Images were acquired every 10 s for 10 min. The time-lapse was started moments after the tissue was collected in frame after the cut. The Slice shown in the video is the same as in *Figure 4c–e*.

DOI: https://doi.org/10.7554/eLife.45019.012

exchanging of nearest neighbors over the course of tens to hundreds of minutes and shows that at the timescale of folding the EGL is more fluid-like than a solid epithelial layer (*Figure 5* and *Videos 3* and *4*).

## Multi-phase wrinkling model simulates cerebellar shape change during folding initiation

We recently developed a model for folding from differentially expanding bi-layer tissues that takes into account the out-of-phase thickness of the outer layer of several systems and possible contribution of radial mechanical constraints present in neurological tissue (*Engstrom et al., 2018*). We applied the model here to the initiation of cerebellar folding based on five primary assumptions. First, the core is an incompressible material (μ) as indicated by the bulk modulus measurements. Second the outer layer, that is the EGL, expands uniformly ($k_t$) as shown by the proliferation rate. Third, the EGL is assumed to be a fluid-like material as demonstrated by the live-imaging of neighbor exchanges. Fourth, there is an elastic component radially to the entire cerebellum ($k_r$), seen in the cutting and relaxation experiment and possibly mediated by

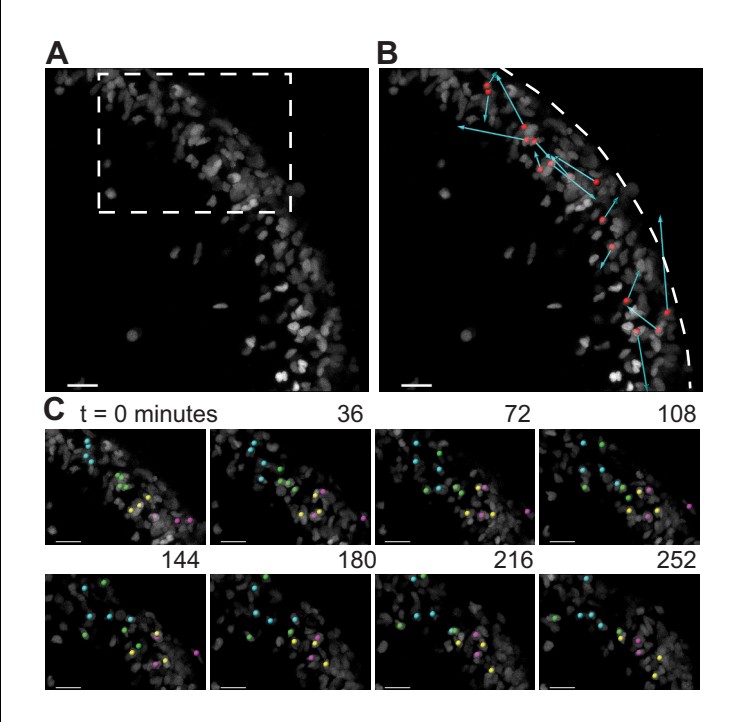

**Figure 5.** During folding initiation, cells within the EGL are motile and undergo rearrangement. (**A**) Image of E16.5 live cerebellar slice (Atoh1-creER/+R26^{Ai75/+}) showing scattered labeling within the EGL. (**B**) Red dots indicate starting position, displacement arrows show final position of marked cells after 5 hr. White dashed line indicates outer edge of EGL. (**C**) Still images from time-lapse, inset above. Cells tracked and marked with colored spheres exchange nearest neighbors over a time-scale of tens of minutes. Scale bars are 20 μm.
DOI: https://doi.org/10.7554/eLife.45019.013

radial glia. Fifth, the EGL is constrained towards a uniform thickness ($\beta$), possibly by Bergmann glia fibers spanning the EGL. Given the interplay between incompressible material, compressible fibrous material, and a proliferating non-elastic EGL, this model is multi-phase.

An energy functional parameterized by both the inner and outer boundary of the EGL and incorporating the above five assumptions into three dimensionless parameters ($\mu/k_r$, $k_r/k_t$, $k_t/\beta$) is minimized to yield an equation for a driven harmonic oscillator resulting in sinusoidal shapes for both the inner and outer boundary of the EGL given an initial elliptical shape. In contrast with the elastic bilayer wrinkling model, EGL thickness oscillations are found to be out-of-phase with the surface height (radius) oscillations when $0 < \mu/k_r < 1$. Additionally, the model predicts that the ratio of the measured surface height amplitude ($A_r$) and the EGL thickness amplitude ($A_t$) is given by

$$\frac{A_r}{A_t} = \frac{\frac{\mu}{k_r}}{1 - \frac{\mu}{k_r}} \ ,$$

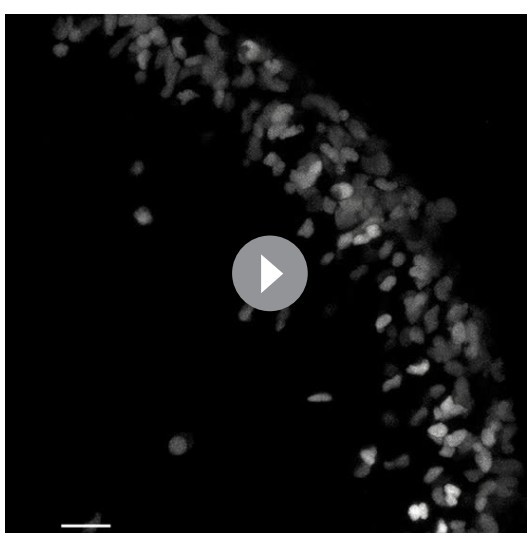

**Video 3.** Live slice imaging reveals fluid-like behavior in the EGL. Time-lapse video shows dynamic cell rearrangement of progenitors within the EGL. Image stacks were acquired every ~3.5 min for 5 hr. Cells undergo mixing and neighbor exchange in the tens to hundreds of minutes.
DOI: https://doi.org/10.7554/eLife.45019.014

which need not be $\gg 1$ as is typical of elastic bilayer wrinkling, and the number of initial folds at E16.5 is determined by

$$n = \sqrt{\frac{k_t}{\beta}} \sqrt{1 + \frac{\frac{\mu}{k_t}}{1 - \frac{\mu}{k_r}}}$$

Note that in contrast with elastic wrinkling, the number of initial folds does not depend on the thickness (a length scale) of the EGL, but only on material properties.

Given that our tissue cutting and relaxation experiment revealed circumferential tension in the cerebellum at folding initiation (*Figure 2f–h*, *Video 1*), we returned to the mathematics and found a previously unrealized geometric relationship in the circular limit of the model that in fact assumes circumferential tension in addition to the previously discussed radial tension given that the perimeter of a circle is determined by its radius.

To rigorously test the shape prediction of the model, we first constrained 3 of the 5 parameters for a circular version of the model by using both the thickness amplitude, and average thickness of the EGL, as measured at E16.5, and the number of initial folds. Secondly, the parameter $\mu/k_r$ (denoted as $\varepsilon$) was assumed to scale linearly with time. Together, this allowed for the generation of shape predictions at later developmental stages (E17.5 and E18.5) from the E16.5 starting approximation. Solving the analytical model as constrained by our measured embryonic data we found that it closely approximates the phase and amplitude behavior of EGL thickness and radius oscillations from E16.5 through E18.5. (*Figure 6a–c*). However, the model is not able to produce self-contacting folds or hierarchical folding, both of which are seen in the cerebellum at later stages.

## Hierarchical folding involves differential growth

The cerebellum has hierarchical folding in which the initial folds become subdivided. Given that ACs hold their position during development and compartmentalize granule cells within lobules of the EGL (*Legué et al., 2015*) we reasoned that the ACs could be acting as mechanical boundaries enabling similar mechanics to drive the secondary folding. To test this possibility we measured the expansion of the EGL and the core of the individual lobule regions from E18.5 to P3. We found that indeed in the lobule regions that undergo folding there is a temporal correlation between when the onset of sub-folding and differential expansion occur (*Figure 7a–d*). In contrast, the region (L45) that does not fold during the same time period has a different, more rectangular shape, and the ratio of EGL growth to core growth is proportional for a rectangle during the time measured (*Figure 7*).

## Discussion

Here we have provided experimental evidence that cerebellar folding emerges without obvious pre-patterning. Additionally, the outer layer has fluid-like properties and expands uniformly, and the growth creates a differential expansion between the outer layer and the core. Thus, traditional morphometric cellular behaviors such as changes in cell shape, size and proliferation do not direct where cerebellar folding initiates. Furthermore, our developmental interrogation revealed tissue moduli, mechanical constraints, and emergent thickness variations in the EGL that are fundamentally inconsistent with traditional elastic bilayer wrinkling models. Therefore our results call for a new understanding of brain folding.

By applying a multi-phase model constrained by our measured data we were able to capture the correct shape variations and number of folds at the onset of folding. Our new framework accounts for: the rapidly expanding fluid EGL, whose thickness is proposed to be regulated by Bergmann glial fibers, the slower growing incompressible core, and fibrous material in the form of glial fibers and possibly axons as well as the meninges that potentially provide radial and circumferential tension (*Figure 8*). This multi-phase model of folding makes many new predictions. One such prediction is that adjusting the amount of tension spanning the cerebellum will change the degree of folding. Indeed, alterations of the cells that likely create tension-based forces could explain the dramatically disrupted folding seen in mouse mutants in which radial glia do not produce Bergmann glia (*Li et al., 2014*). Without Bergmann glia, the EGL would be expected to not form a layer with regular thickness and it should be more sensitive to variations in radial glial tension. Consistent with this prediction, mutants without Bergmann glia have more localized and less regular folds (*Li et al., 2014*)

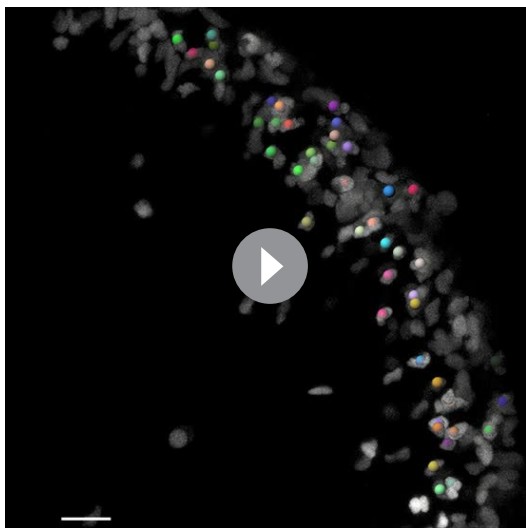

**Video 4.** Live slice imaging reveals fluid-like behavior in the EGL. Time-lapse shown in *Video 3*, with a subset of cells tracked through time and their positions marked with colored spheres.
DOI: https://doi.org/10.7554/eLife.45019.015

Our combination of experimental studies and modeling thus provide new insights into cerebellar folding, including an underappreciated role for tension.

Under the new framework revealed by our measurements made in the developing mouse cerebellum, to approximate the observed shape changes in the murine cerebellum from E17.5 to E18.5 the ratio of the core stiffness over the radial tension must increase. Yet, the measured bulk modulus of the core shows no increase during development. Therefore a second prediction is that radial tension must decrease during development. While the cerebellum is crossed by many fibers at folding initiation, radial glial fibers are an attractive candidate to mediate this change in radial tension (*Sillitoe and Joyner, 2007*; *Rahimi-Balaei et al., 2015*). First, they span from the VZ to the surface of the cerebellum at E16.5. Additionally, during folding initiation the radial glia undergo a transition into Bergmann glia where they release their basal connection to the VZ and the cell body migrates towards the surface (*Mota and Herculano-Houzel, 2015*). This transition could lead to a reduction in the global radial tension and thus would be consistent with our model prediction.

The mechanics underlying hierarchical folding remain an open challenge. However, our developmental data may provide a way forward. As ACs maintain their spatial positions, and as they compartmentalize granule cells within the EGL into the lobule regions (*Legué et al., 2015*), we propose that they create fixed mechanical boundaries that divide the cerebellum into self-similar domains. These domains, with their similar physical properties to the initial unfolded cerebellum, can then undergo additional folding. Furthermore since ACs compartmentalize granule cells within the lobule regions, once separated the lobule regions can develop distinct characteristics, like the observed differential proliferation rates at E18.5. We speculate, therefore, that the folding patterns seen across

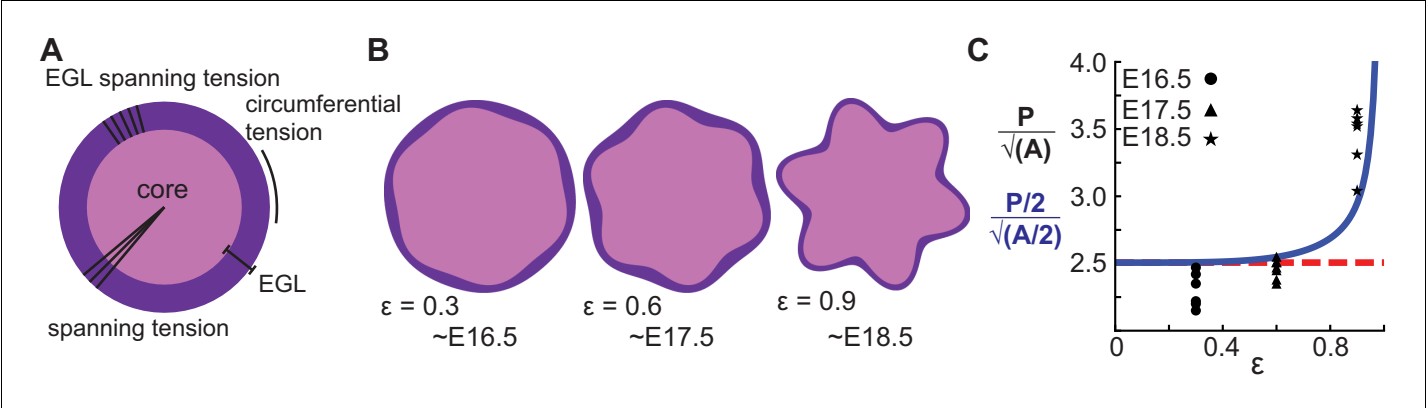

**Figure 6.** A multiphase model with radial and circumferential constraints and liquid-fibrous EGL composition approximates evolution of cerebellar shape. (A) Schematic of multiphase model showing types of tension. (B) Thickness variations that arise concomitant with folding approximate those seen in the cerebellum. (C) Shape factor analysis: model for semicircle (red), multi-phase model shape prediction (blue) and actual shape measured from sections (black). Assumed linear relationship between $\varepsilon$ and time, $\varepsilon(T) = 0.3(T - 15.5)$.
DOI: https://doi.org/10.7554/eLife.45019.016

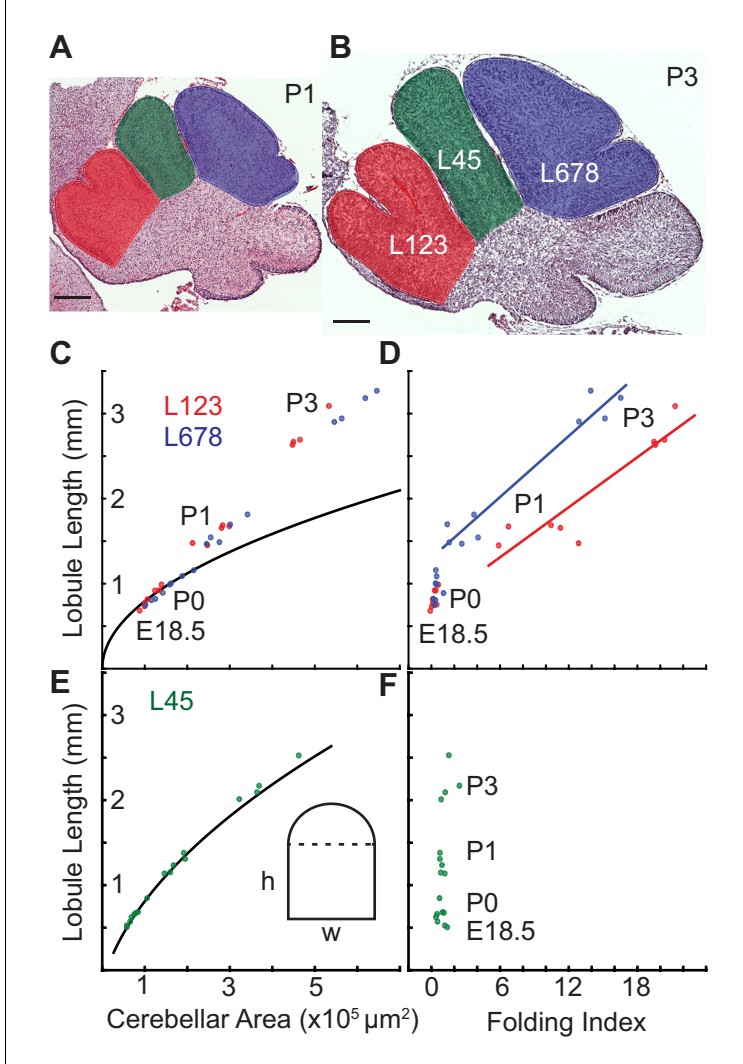

**Figure 7.** Differential expansion correlates with progressive subfolding of initial folds. (**A,B**) H and E stained midline sagittal sections of FVB/N cerebella at P1 and P3 with three lobule regions highlighted in red (L123), green (**L45**), and blue (L678). (**C**) Expansion of lobule length and lobule area for L123 and L678 approximate the proportional expansion of a semi-circle (curve) in both regions at E18.5 and P0. After P0 the EGL expansion in both regions increases more than the underlying area creating differential expansion. (**D**) Folding initiates during regional differential expansion. (**E**) The expansion in length and area of L45 is proportional to a columnar shape (curve and inset figure) from E18.5 to P3. (**F**) the L45 region remains unfolded through P3. Scale bars: 200 μm.
DOI: https://doi.org/10.7554/eLife.45019.017

cerebella in different species evolved by adjustment of global as well as regional levels of differential expansion and tension which ultimately mold the functionality of the cerebellum.

Finally it is interesting to note the similarities and differences between the developing cerebellum and the cerebral cortex. Radial glia span the entire cerebral cortex just as in the cerebellum (*Götz et al., 2002*). Furthermore, species with folded cerebrums have evolved outer radial glial cells for which the cell body leaves the ventricular zone to become positioned near the surface while retaining fibers anchored on the surface, similar to Bergmann glia in the cerebellum (*Leung and Li, 2018*; *Reillo et al., 2011*). While we have emphasized the notion of tension via glial fibers in the developing cerebellum, axonal tension has been discussed in the context of shaping the developing cerebrum (*Van Essen, 1997*). Tissue cutting in the cerebral cortex of ferrets has revealed a similar tension pattern during folding as we found in the cerebellum (*Xu et al., 2010*). We therefore submit

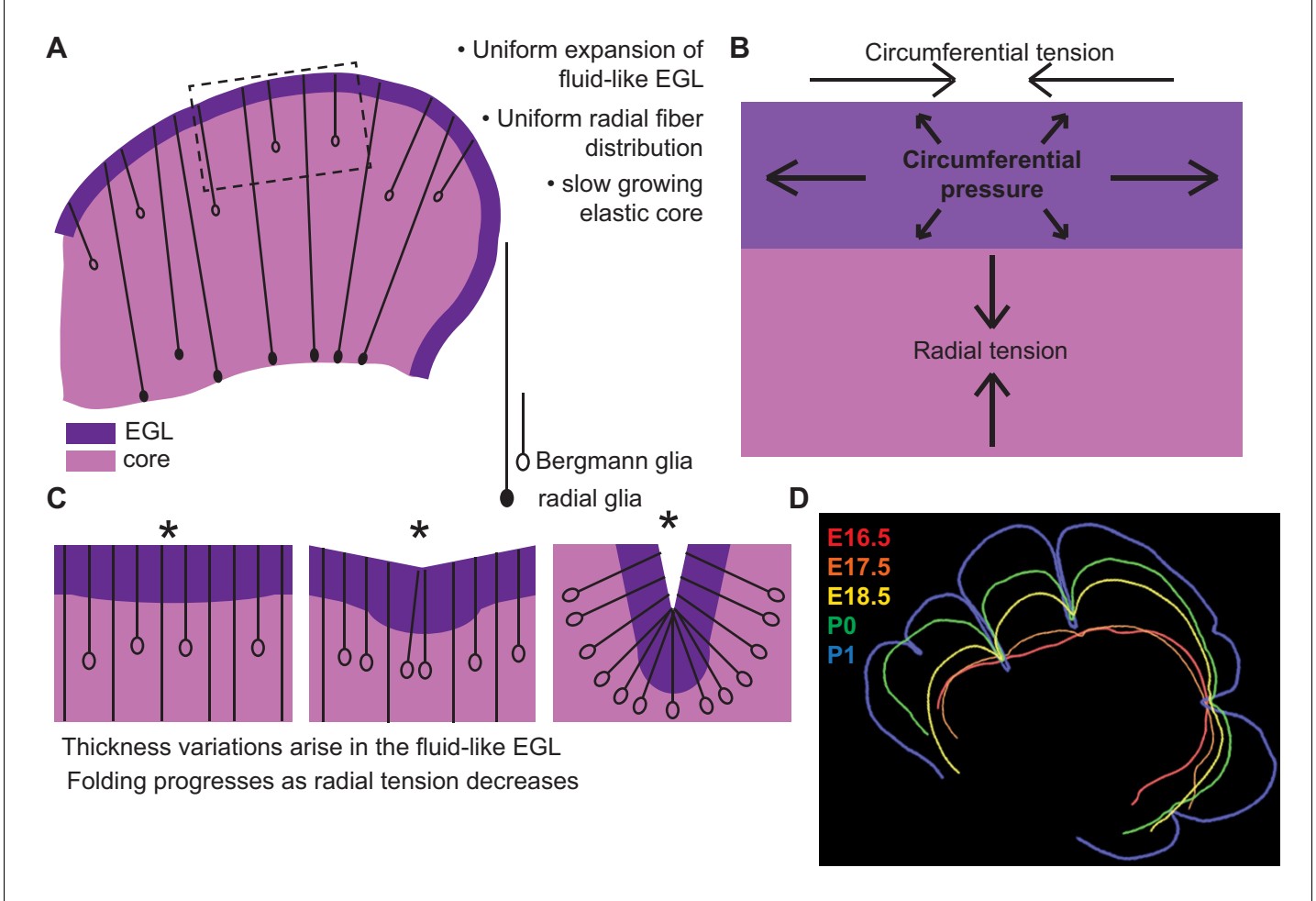

**Figure 8.** Uniform cell behaviors within a fluid-like EGL create differential expansion based folding approximated by a multiphase model with radial and circumferential tension. (**A**) Cartoon of E16.5 cerebellum showing the EGL (dark purple) overlying the in incompressible core (light purple) with fibers (lines) of radial glia (closed ovals) and Bergmann glia (open ovals) crossing the cerebellum and the EGL. (**B**) Map of stress within the cerebellum at the onset of foliation. (**C**) Schematics showing that an AC is first detected as a regional inward thickening of the EGL (left). The constraining tensions shape the fluid-like EGL such that the EGL becomes thicker at the AC (middle). As radial glia transition to Bergmann glia, modeling predicts a reduction in radial tension (right). (**D**) Since ACs hold their position in space, and compartmentalize the cells within the EGL, we propose that they behave as mechanical boundaries allowing local domains of differential expansion to arise and progressive folding to occur.
DOI: https://doi.org/10.7554/eLife.45019.018

that our work calls for a revival of the notion of how tension affects the shape of the developing cerebrum.

Unlike the cerebellum, the cerebral cortex is not divided into a simple bilayer system. However, outer radial glial cells proliferate, much like the GCPs of the EGL, to drive the expansion of the outer regions of the cerebral cortex around the time of initiation of folding (*Hansen et al., 2010*; *Heng et al., 2017*; *Nowakowski et al., 2016*). Moving the zone of proliferation out from the VZ gives more space for the increased proliferation required in folding systems. The cerebellum, housing 80% of the neurons in the human brain may be an extreme example requiring the region of proliferation to be completely on the outer surface (*Andersen et al., 1992*). Constraining models of folding of different brain regions with developmental data will bring about a more accurate quantitative understanding of the shaping of the developing brain.

# Materials and methods

**Key resources table**

| Reagent type (species) or resource | Designation | Source or reference | Identifiers | Additional information |
|---|---|---|---|---|
| Genetic reagent (M.Musculus) | FVB/N | Jax Mice, The Jackson Laboratory | Jackson Labs Stock number: 001800 | |
| Genetic reagent (M.Musculus) | Atoh1-CreER | *Machold and Fishell, 2005* | Jackson Labs Stock number: 007684 | |
| Genetic reagent (M.Musculus) | Nestin-CreER | *Imayoshi et al., 2006* | | |
| Genetic reagent (M.Musculus) | $Rosa26^{MTMG}$ | *Muzumdar et al., 2007* | Jackson labs Stock number: 007676 | |
| Genetic reagent (M.Musculus) | $Rosa26^{Ai75}$ | *Daigle et al., 2018* | Jackson labs Stock number: 025106 | |
| Antibody | mouse monoclonal anti-P27 | BD Pharmingen | 610241 | Dilution 1:500 |
| Antibody | rabbit polyclonal anti-GFP | Life Technologies | A11122 | Dilution 1:500 |
| Antibody | rat monoclonal anti-GFP | Nacalai Tesque | 04404–84 | Dilution 1:500 |
| Commercial Assay or Kit | EDU | Invitrogen | C10340 | |

## Animals

All experiments were performed following protocols approved by Memorial Sloan Kettering Cancer Center's Institutional Animal Care and Use Committee. The inbred FVB/N stain was used for all proliferation rate, area, length, and expansion rate measurements. *Atoh1-CreER* (*Machold and Fishell, 2005*), *Nestin-CreER* (*Imayoshi et al., 2006*), $Rosa26^{MTMG}$ (*Muzumdar et al., 2007*), $Rosa26^{Ai75}$ (*Daigle et al., 2018*) were used to quantify cell shape and size as well as fiber distribution and were maintained on the outbred Swiss Webster background. The Swiss Webster strain was used for scanning acoustic microscopy. Both sexes were used for the analysis. Animals were kept on a 12 hr light/dark cycle and food and water were supplied ad libitum. All experiments were performed following protocols approved by Memorial Sloan Kettering Cancer Center's Institutional Animal Care and Use Committee.

The appearance of a vaginal plug set noon of the day as Embryonic day 0.5 (E0.5). All animals were collected within two hours of noon on the day of collection. Tamoxifen (Tm, Sigma-Aldrich) was dissolved in corn oil (Sigma-Aldrich) at 20 mg/mL. Pregnant females carrying litters with *Atoh1-CreER/+;R26$^{MTMG/MTMG}$* or *NestinCER/+;R26$^{MTMG/MTMG}$* embryos were given one 20 µg/g dose of TM via subcutaneous injection two days prior to analysis. 25 µg/g of 5-ethynyl-2-deoxyruidine (EDU; Invitrogen) was administered via subcutaneous injection one hour prior to collection.

## Tissue processing, immunohistochemistry, and imaging

For embryonic stages heads were fixed in 4% paraformaldehyde overnight at 4°C. For postnatal animals, the brain was dissected out first before fixation. Tissues were stored in 30% sucrose. For all proliferation, area, length, and thickness measurements brains were embedded in optimal cutting temperature (OCT) compound. Parasagittal sections were collect with a Leica cryostat (CM3050s) at 10 µm.

Prior to IHC, EdU was detected using a commercial kit (Invitrogen, C10340). Following EdU reaction the following primary antibodies were used either overnight at 4°C or 4 hr at room temperature: mouse anti-P27 (BD Pharmingen, 610241), rabbit anti-GFP (Life Technologies, A11122), rat anti-GFP (Nacalai Tesque, 04404–84). All antibodies were diluted to 1:500 in 2% milk (American Bioanalytical) and 0.2% Triton X-100 (Fisher Scientific). Alexa Fluor secondary antibodies (1:500; Invitrogen) were used: Alexa Fluor 488 donkey anti-rabbit, A21206, Alexa Fluor 488 donkey anti-rat, A21208, Alexa

Fluor 488 donkey anti-mouse, A21202, Alexa Fluor 647 donkey anti-mouse, A31571. EdU was detected using a commercial kit (Invitrogen, C10340).

For cell size, shape and fiber density analysis 60 µm parasagittal sections were collected on a Leica vibratome (VT100S). Primary and secondary antibodies were diluted 1:500 in 2% milk and incubated overnight at 4°C.

For scanning acoustic microscopy brains were processed for paraffin embedding and parasagittal sections of 10 µm thick were collected on a microtome (Leica RM2255). Structured illumination and confocal Imaging was done with Zeiss Observer Z.1 with Apatome or Zeiss LSM 880 respectively.

## Quantification of proliferation, length, area, folding index and thickness

Measurements for all analysis were taken from the three most midline sagittal sections and averaged. The most midline section was determined by dividing the distance in half between the lateral edges where the third ventrical and the mesencephalic vesicle are no longer connected. Quantifications were made using Imaris (Bitplane) and MATLAB (Mathworks) software.

EGL Proliferation rate was calculated as EDU+/(Dapi+;P27-) cells. All cells were counted within the lobule region to the midpoint of the Anchoring Centers. For proliferation measurements through the ACs and the surrounding EGL at E16.5 and E17.5 a 50 µm window measured from the outer surface of the EGL was centered at the AC. The measuring window was centered at every 25 µm anterior and posterior to the EGL for a total distance of 250 µm anterior and posterior to the AC. At E18.5, when the AC is fully formed, everything proximal to the centroid of the cerebellum under the midpoint of the AC was counted as the AC. Non-overlapping regions of 50 µm also were measured in either direction for a total of 200 µM anterior and posterior to the AC. Proliferation was measured in three cerebella at E16.5 and E17.5 and in four cerebella at E18.5.

EGL length was measured from the outer surface of the EGL following the curvature of the EGL. Cerebellar area was calculated as the area within the outer surface of the EGL and the ventricular zone. A short strait edge was made perpendicular to the ventricular zone to close the area back upon to the anterior end of the EGL. The convex curvature of the cerebellum was measured by following only the positive curvature of the EGL. The folding index was determined as FI = 1 - (Positive curvature/EGL length). Data collected for E16.5, E17.5, E18.5 and P0 came from 6,8,7, and 9 cerebella respectively.

EGL thickness was measured by defining the outer and inner curvature of the EGL. The shortest distance lines were drawn to the outer curvature from discrete points distributed at every 12.5 µm along the inner curvature of the EGL. Nine ACs and surrounding regions from five cerebella were quantified at E16.5 and 13 ACs from five cerebellar were analyzed at E17.5. At E18.5 six ACs from two cerebellar were quantified.

## Quantification of cell shape

Midline sections were imaged with a Zeiss LSM 880. Serial images were taken to cover the entire EGL of lobule regions L123, L45, and L678 and the ACs. Manual cell masks were created with Imaris software defining the curvature at every z-slice. Every cell that was completely included in the imaging window and that was distinguishable from surrounding cells was counted to reduce sampling bias. Cells from three brains were measured at each stage for a total of 131 at E16.5 and 201 at E18.5. Shape was defined via sphericity, which is the surface area of a sphere having the same volume as the cell of interest divided by the surface area of the cell of interest.

## Quantification of fibers within the EGL

Midline sections were imaged with a Zeiss LSM 880. Image tiling was used to cover the EGL. Using Python, a fourth or fifth order polynomial was fitted to the outer edge of the EGL in each image, and five scan lines were positioned at 12.2 µm intervals beneath the surface, and parallel to it. A bin width of 50 µm as measured along the polynomial contour was centered at the AC. Bins of equal distance were extended both anteriorly and posteriorly. Staining intensity was counted along each scan line at every z-slice of the confocal stack. Each image was normalized to the mean intensity and smoothed with a Gaussian filter. Peak counting was done using minimum and maximum filters, keeping neighborhood size and threshold parameters constant for all images. The results from the five scan lines were averaged.

### Tissue cutting

Live cerebella of E16.5 FVB/N mice were collected in dissection buffer as previously described (*Wojcinski et al., 2017*) and embedded in low-melting point agarose (Invitrogen). Sagittal slices at a thickness of 250 μm were collected. Slices were removed from the agarose and place in petri-dishes coated with Poly(2-hydroxyethyl methacrylate)(Sigma-Aldrich). Tissue cuts (eight horizontal, 10 radial) were made with a 30° Premier Edge stab knife (Oasis Medical). Slices were allowed to relax for 10 min. Time-lapse images were acquired on a Leica MZ75 dissection scope.

### Live imaging analysis

Live cerebella of E16.5 Atoh1-CreER/+; R26Ai75/+mice were collected and slices of a thickness of 250 μm were cultured on Millicell cell culture inserts (Millipore) in glass bottom plates (Matek) as previously described (*Wojcinski et al., 2017*). Image stacks were acquired on a Zeiss LSM 880 at intervals of around 3.5 min for up to 5 hr. Cell positions were tracked using Imaris (Bitplane) software. Three time-lapses were analzyed.

### Scanning acoustic microscopy

Mechanical tissue properties were analyzed using a 250 MHz Scanning Acoustic Microscope (SAM), described previously (*Rohrbach et al., 2015*; *Rohrbach et al., 2018*; *Rohrbach and Mamou, 2018*). Briefly, 12 μm paraffin sections of mouse embryonic brains were de-parafinized, hydrated in de-ionized water and raster scanned (2 μm steps in both direction) on the SAM to acquire radio-frequency (RF) ultrasound data. At each scan location, signal processing was performed to compute the amplitude, sample thickness, speed of sound, acoustic impedance, attenuation, bulk modulus, and mass density (*Rohrbach and Mamou, 2018*). Two-dimensional maps of tissues properties were formed using the values obtained at each scan location. Bulk modulus was computed from the product of the acoustic impedance and the speed of sound. Co-registered histology and SAM amplitude images were used to identify regions-of-interest (ROIs) corresponding to the EGL layer and underlying core of the cerebellum in each sample. Bulk modulus was analyzed as a measure of tissue stiffness: ROI measurements were acquired from 3 sections from three embryos at each developmental stage.

### Finite element simulations

The wrinkle of a circular bilayer structure in *Figure 3a* was simulated with commercial software ABAQUS. Both film and substrate were modeled as incompressible neo-Hookean materials. The ratio between shear moduli of the film and substrate was 50 and the initial radius of the simulated structure was 16 times that of the film thickness. The differential growth of the EGL and core was modeled by an isotropic expansion of the film in the bilayer structure.

To test the elastic wrinkling model, we conducted finite element (FE) simulations for bilayer structures with a film bonded on a substrate, which represents the EGL layer and core structure, respectively. The structures were assumed to be under 2D plane strain deformation to mimic the quasi-2D nature of cerebellum wrinkles. Neo-Hookean model was adopted to describe the elastic properties of both film and substrate, whose strain energy can be expressed as

$$U = \frac{1}{2}\mu(I_1 - 3)$$

where $\mu$ is the shear modulus and $I_1$ represents the first invariant of the right Cauchy-Green strain tensor. The Poisson's ratios for the film and substrate were set to be 0.5, based on experimental observations that the bulk modulus of EGL and core are in the order of GPa, much larger than the shear modulus of soft tissues ($\sim$ kPa).

We carried out FE simulations through commercial software ABAQUS. A second order 6 node hybrid element (CPE6MH) was utilized to discretize the film and substrate. Very fine FE meshes were used to make sure the results independent of mesh size. To incorporate differential growth in real EGL layer and core, an isotropic growth deformation tension was applied to the modeled film by decoupling the deformation tenor $\boldsymbol{F}$ into elastic deformation part $\boldsymbol{A}$ and growth part $\boldsymbol{G}$.

$$F = A \cdot G$$

For simplicity, we assume the growth part is isotropic and controlled by a scalar variable $g$

$$G = g \begin{bmatrix} 1 & 0 & 0 \\ 0 & 1 & 0 \\ 0 & 0 & 1 \end{bmatrix}$$

where $g > 1$ represents a faster growth in EGL than the core. To trigger instabilities in numerical simulations, random perturbations (e.g., White Gaussian noise with $0.001t$ mean magnitude) were applied to the nodal positions at the top surface of the film and the interface between the film and substrate.

To qualitatively understand the cut experiments we ran a FE simulation of a pre-cut circular bilayer structure and then assigned swelling strain to the film. This neglected the dynamical process in the real cut experiments and only focused on the final equilibrium of the cerebellum after long time relaxation. All the simulation parameters were the same as those in the wrinkling simulation. The initial cut length $a$ is equal to 8 t. The minimum in-plane principal stress corresponds to the hoop stress in the film.

## Details of multi-phase model as applied to initiation of cerebellar folds

For a full treatment of the mathematics please see *Engstrom et al. (2018)*.

We, formulated a two-dimensional model based on the parameters of a midsagittal section of the cerebellum. The distance of the outer edge of the EGL and, hence, the outer edge of the cerebellum from the center of the cerebellum was defined as $r(\theta)$ with $\theta$ as the angular coordinate. We assumed that $r(\theta)$ was single-valued. The thickness of the EGL was defined as $t(\theta)$. See model schematic in.

Taking into account the four assumptions discussed in the main text, we constructed the following energy functional to be minimized

$$E\left[r, t, \frac{dt}{d\theta}\right] = \int d\theta \left\{ k_r(r - r_0)^2 - k_t(t - t_0)^2 + \beta\left(\frac{dt}{d\theta}^2\right) \right\},$$

with $k_r$ as the stiffness modulus (a spring constant in one-dimension) of the radial glial fibers and the pial surface contained in the meninges surrounding the cerebellum since the cerebellar radius is proportional to its perimeter, $r_0$ as the preferred radius of the cerebellum, $k_t$ denoting a growth potential due to cell proliferation, $t_0$ as thickness of the EGL (cortex), and, $\beta$ quantified the mechanical resistance to changing the thickness of the EGL. Given our first assumption of an incompressible cerebellar core, we imposed the constraint

$$\frac{1}{2} \int d\theta (r - t)^2 = A_0,$$

with $A_0$ as a preferred cerebellar area. We applied the variational principle to minimize the energy functional subject to the core constraint, that is

$$\delta\left(E - \mu \int d\theta\, (r - t)^2\right) = 0,$$

where $\mu$ is a Lagrange multiplier. Assuming the preferred radius of the cerebellum is constant and the thickness of the EGL/cortex is also constant, then the preferred cerebellar shape was a circle and the EGL an annulus.

The variational analysis yielded the following equation of shape for $t(\theta)$;

$$\frac{d^2 t}{d\theta^2} + q^2\, t(\theta) = \frac{k_t}{\beta}\left(t_0 + \frac{\frac{\mu r_0}{k_t}}{1 - \frac{\mu}{k_r}}\right),$$

with $q^2 = \frac{k_t}{\beta}\left(1 + \frac{\frac{\mu}{k_t}}{1 - \frac{\mu}{k_r}}\right)$. The solution to the equation of shape was

$$t(\theta) = A_t \sin(q\theta + \phi) + C_1(r_0, t_0, k_r, k_t, \mu),$$

with $C_1$ independent of $\theta$ and $A_t = \sqrt{2}\left(1 - \frac{\mu}{k_r}\right)\sqrt{\frac{A_0}{\pi} - C_2(r_0, t_0, k_r, k_t, \mu)}$ such that $A_0 > \pi C_2$. There was an additional equation of shape for $r(\theta)$ from the variational principle that depended on $t(\theta)$ and so was determined

$$r(\theta) = -\frac{\frac{\mu}{k_r}}{1 - \frac{\mu}{k_r}} A_t \sin(q\theta + \phi) + C_3(r_0, k_r, \mu).$$

We used the measured data at E16.5 to set the parameters to make predictions for the shape of both the EGL and core (and so the relationship between the two) at later times. Because we are primarily interested in shape changes, rather than size changes, a nondimensionalized model solution was used, that is we chose units where $r_0 = 1$. This reduces the total number of parameters specifying the model to five dimensionless parameters. Plots assumed a circular preferred shape, and with other parameters as follows: $\epsilon = \mu/k_r$ is shown in **Figure 6b,c**, $c = k_r/k_t = 0.06/\epsilon$, $A_t/r_0 = \epsilon/9.6$, $t_0/r_0 = \epsilon/4.8$, and $q = 6$. Note that for $\epsilon = 0.3$, these parameters are numerically consistent with our E16.5 measurements: $A_t/t_0 = 0.5$ and $r_0/t_0 = 16$, as well as the observed number of invaginations in the half circle: $q/2 = 3$. All of these parameters are either constant or depend on the time-like parameter $\epsilon$. One of these dependencies has a functional form that is physically justifiable ($A_t \sim \epsilon$), another has a form that is biologically justifiable ($c \sim 1/\epsilon$), owing to the decrease in the number of radial glia over time.

We defined a dimensionless 'shape factor' as half of the perimeter divided by the square root of half of the area as appropriate for a semi-circle. To compare the model's predictive deviation of this quantity form the semi-circular value we assumed a linear relationship between $\epsilon$ and time T measured in embryonic days: $\epsilon(T) = 0.3(T-15.5)$.

## Statistical analyses

Statistical analyses were performed using Matlab software. Significance was determined at $p < 0.05$. Two-way ANOVA was used for proliferation analysis as two variables were tracked, mouse and region. Cell shape, volume, fiber distribution, EGL thickness and bulk modulus were run under a standard ANOVA. After ANOVA analysis a multiple comparison was run with Tukey's honestly significant difference criterion. F-test for variance and two-tailed student's paired t-test were used for slice cutting and relaxation quantifications. The degrees of freedom, where appropriate, and P values are given in the figure legends. All error bars are standard deviations. No statistical methods were used to predetermine the sample sizes. We used sample sizes aligned with the standard in the field. No randomization was used nor was data collection or analysis performed blind.

## Acknowledgements

We are grateful to Anna-Katerina Hadjantonakis and Songhai Shi for use of Imaris software. We thank Jennifer Zallen and Anna-Katerina Hadjantonakis for experimental advice, Nathanael Kim for help with the acoustic microscopy and Professor Tadashi Yamagushi for his support for MO's visit to New York. We appreciate the discussions we have had with Alexandre Wojcinski and the entire Joyner Laboratory, and the administrative support from Cara Monaco.

## Additional information

### Funding

| Funder | Grant reference number | Author |
| --- | --- | --- |
| National Institute of Neurological Disorders and Stroke | Postdoctoral Training Fellowship: F32NS086163 | Andrew K Lawton |
| National Institute of Biomedical Imaging and Bioengineering | R21EB016117 | Jonathan Mamou |

| | | |
|---|---|---|
| National Science Foundation | Comet Cluster in the Extreme Science and Engineering Discovery Environment: TG-MSS170004 | Teng Zhang |
| National Science Foundation | NSF-DMR-CMMT 1507938 | J M Schwarz |
| National Science Foundation | NSF-PHY-PoLS 1607416 | J M Schwarz |
| National Institute of Mental Health | R37MH085726 | Alexandra L Joyner |
| National Cancer Institute | Cancer Center Support Grant: P30 CA008748-48 | Alexandra L Joyner |
| National Institute of Mental Health | R01NS092096 | Alexandra L Joyner |

The funders had no role in study design, data collection and interpretation, or the decision to submit the work for publication.

## Author contributions

Andrew K Lawton, Conceptualization, Formal analysis, Funding acquisition, Validation, Investigation, Visualization, Methodology, Writing—original draft, Project administration, Writing—review and editing; Tyler Engstrom, Conceptualization, Formal analysis, Validation, Investigation, Visualization, Methodology, Writing—review and editing; Daniel Rohrbach, Formal analysis, Supervision, Investigation, Methodology, Writing—review and editing; Masaaki Omura, Formal analysis, Investigation; Daniel H Turnbull, Supervision, Project administration, Writing—review and editing; Jonathan Mamou, Resources, Supervision, Project administration, Writing—review and editing; Teng Zhang, Formal analysis, Funding acquisition, Validation, Investigation, Visualization, Methodology, Writing—review and editing; J M Schwarz, Conceptualization, Supervision, Funding acquisition, Investigation, Methodology, Project administration, Writing—review and editing; Alexandra L Joyner, Conceptualization, Supervision, Funding acquisition, Writing—original draft, Project administration, Writing—review and editing

## Author ORCIDs

Andrew K Lawton (iD) https://orcid.org/0000-0001-8633-6637
Alexandra L Joyner (iD) http://orcid.org/0000-0001-7090-9605

## Ethics

Animal experimentation: All experiments were performed following protocols approved by Memorial Sloan Kettering Cancer Center's Institutional Animal Care and Use Committee. Protocol number: 07-01-001 Mouse Developmental Genetics

## Decision letter and Author response

Decision letter https://doi.org/10.7554/eLife.45019.021
Author response https://doi.org/10.7554/eLife.45019.022

# Additional files

## Supplementary files

• Transparent reporting form
DOI: https://doi.org/10.7554/eLife.45019.019

## Data availability

All data is presented in the manuscript and supporting files.

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
