## [Decision Letter]

Thank you for submitting your article "Cerebellar folding is initiated by mechanical constraints on a fluid-like layer without a cellular pre-pattern" for consideration by *eLife*. Your article has been reviewed by two peer reviewers, and the evaluation has been overseen by a Reviewing Editor and Marianne Bronner as the Senior Editor. The reviewers have opted to remain anonymous.

The reviewers have discussed the reviews with one another and the Reviewing Editor has drafted this decision to help you prepare a revised submission.

Summary:

Both reviewers thought that your study of cerebellar folding in developing murine embryo addresses an important outstanding question in the field of developmental neuroscience. They considered your studies of tissue dynamics and cellular behavior analyses, as well as mathematical simulations as comprehensive and compelling. The observations that during the cerebellar folding initiation differential expansion is due to the outer layer of proliferating progenitors expanding faster than the tissue core and that the folding occurs without an obvious cellular pre-pattern, are particularly interesting. These and other observations challenge the existing models and allow putting forward the multi-phase wrinkling model. The manuscript should be of broad interest to developmental biologists, cellular biologist, neuroscientists and applied mathematicians.

Essential revisions:

One of the reviewers is concerned that in the model the missing two parameters μ/k_r_ are constrained to scale linearly in time. There must be a second constraint to fix two parameters. The motivation behind linear scaling and the full choice of parameters should be more transparent.

*Reviewer #1:*

Many models were proposed to explain how brain folds. The elastic wrinkling models based on variations in stiffness, compressive forces, proliferation, and thickness has been used to explain the cerebellar folding. In this manuscript, Joyner and her colleagues challenged the folding models by measuring those values and cellar behaviors (e.g. cell shape, motility, and glial fibers) in mouse cerebella when the folding is initiated, and by mathematical simulation. They demonstrated via experimental data: (1) differential expansion and stiffness between the EGL vs. the core, (2) the EGL thickness at the anchoring centers (ACs) increases but not decreases during the folding initiation, (3) uniform proliferation in the EGL, (4) presence of horizontal and radial mechanical constrains, (5) fluid-like property of the EGL. With the mathematical simulation, they propose a novel multi-phase wrinkling model without introducing a cellular pre-pattern. They also explain the subdivisions after the initial folding (they termed hierarchical folding) when they consider ACs as mechanical boundaries. Although I am not familiar with mathematical simulation, I believe that they show enough evidence supporting their hypothesis. The no pre-pattern model was unexpected. The manuscript provides scientists in the research field of developmental biology and neuroscience with a novel idea for brain folding.

*Reviewer #2:*

The work by Lawton et al. investigates the mechanism behind cerebellar folding by conducting a nicely detailed analysis of the dynamics and mechanical properties of the contributing tissues. What drives brain folding has excited many researchers, developmental biologists, physicists and applied mathematicians alike. Yet, a detailed quantification of the tissue dynamics and tissue properties has so far been lacking. The authors unfold in their very timely contribution how predictions from models guided their quantification of tissue dynamics to step by step rule out any previous models of brain folding and eventually deduct the key ingredients that they consider in a finite element model. They very carefully constrain parameters of their model to then arrive at a very convincing and consistent mechanism of how cerebellar folding unfolds.

My only major concern is that in their model the missing two parameters μ/k_r_ are constrained to scale linearly in time. There must be a second constraint to fix two parameters. The motivation behind linear scaling and the full choice of parameters should be more transparent.

---

## [Author Response]

Essential revisions:One of the reviewers is concerned that in the model the missing two parameters mμ/k_r_ are constrained to scale linearly in time. There must be a second constraint to fix two parameters. The motivation behind linear scaling and the full choice of parameters should be more transparent.

We thank the reviewer for their comment on the parameterization of the model as it gives us a chance to clarify our approach. In the modeling community, it is quickly becoming the norm to reformulate a model in terms of dimensionless quantities, i.e. nondimensionalization. To do so, one works with dimensionless ratios of the parameters, thereby reducing the number of parameters to uncover a smaller set of quantities that the system depends on. In the manuscript, we also work with a nondimensionalized form of the model (i.e., we work in units where r_0_=1), and so only 5 dimensionless parameters are required to completely specify the model. These may be chosen as μ/k_r_, k_r_/k_t_, A_t_/r_0_, t_0_/r_0_, and q, as discussed in the Materials and methods section entitled “Details of Multi-phase model […]”. Thus, the quantity μ/k_r_ should not be thought of as a ratio of two independent parameters, but rather a single dimensionless parameter. We indeed should have been more transparent about using the nondimensionalized version of the model, and we thank the reviewer for pointing out this oversight. The revised manuscript has language added to the section "Details of multi-phase model […]" to state explicitly that the nondimensionalized model is used to make the plots in Figure 6B-C as follows:

“Because we are primarily interested in shape changes, rather than size changes, a nondimensionalized model solution was used, i.e., we chose units where r_0_=1. This reduces the total number of parameters specifying the model to five dimensionless parameters.”

In re-reading this section, we also noticed a typo that may have been the source of some of the confusion. The fifth paragraph said "Figure 5F" where it should say "Figure 6B-C". We apologize for this typo, we have fixed it in the revised manuscript, and hope this clarifies how the parameters in Figure 6B-C are chosen.

Regarding time-dependence, the model parameter μ/k_r_ should be increasing over time because k_r_ is decreasing over time, which we suggest in the Discussion can be attributed to the radial glia transitioning to Bergmann glia (see, e.g., Discussion, third paragraph). We assume linear scaling, because that is the simplest possible kind of scaling. That this is an assumption stated in both the legend to Figure 6 and in the section "Details of multi-phase model […]". Having said that, Figure 6C illustrates that this assumption is not too unreasonable, at least concerning the agreement of the experimental and theoretical shape index.